Enhanced susceptibility to predation in corals of compromised condition

Bright Allan J. 1 2 allan.bright@noaa.gov
Cameron Caitlin M. 1 2
Miller Margaret W. 2
1 Cooperative Institute for Marine and Atmospheric Studies, University of Miami , Miami, FL , USA
2 National Marine Fisheries Service, Southeast Fisheries Science Center , Miami, FL , USA
Esteban María Ángeles
Electronic publication date: 2015 Sep 10
Publication date: 2015
Volume: 3
Electronic Location ID: e1239
Received 2015 Jul 3; Accepted 2015 Aug 22
Copyright year: 2015
License: This is an open access article, free of all copyright, made available under the Creative Commons Public Domain Dedication. This work may be freely reproduced, distributed, transmitted, modified, built upon, or otherwise used by anyone for any lawful purpose.
License URL: https://creativecommons.org/publicdomain/zero/1.0/

Keywords: Coralliophila abbreviata, Acropora cervicornis, Florida Keys, Coral disease, Acroporid, Corallivore

Funding: NOAA Coral Reef Conservation Program This study was supported by the NOAA Coral Reef Conservation Program. The funders had no role in study design, data collection and analysis, decision to publish, or preparation of the manuscript.

==============================
The marine gastropod, Coralliophila abbreviata, is an obligate corallivore that causes substantial mortality in Caribbean Acropora spp. Considering the imperiled status of Acropora cervicornis and A. palmata, a better understanding of ecological interactions resulting in tissue loss may enable more effective conservation strategies. We examined differences in susceptibility of A. cervicornis to C. abbreviata predation based on coral tissue condition. Coral tissue condition was a strong determinant of snail prey choice, with snails preferring A. cervicornis fragments that were diseased or mechanically damaged over healthy fragments. In addition, snails always chose fragments undergoing active predation by another snail, while showing no preference for a non-feeding snail when compared with an undisturbed prey fragment. These results indicate that the condition of A. cervicornis prey influenced foraging behavior of C. abbreviata, creating a potential feedback that may exacerbate damage from predation in coral populations compromised by other types of disturbance.

Introduction

Corallivory is widely understood to have significant, if sometimes underestimated, effects on scleractinian coral populations (Rotjan & Lewis, 2008). Depending on the type and intensity of predation, corallivory may result in positive or negative ecological responses (Cole, Pratchett & Jones, 2008). While moderate predation pressure has sometimes been positively correlated with an increase in species diversity at the community level (Menge & Sutherland, 1976), for an individual coral, predation often equates to partial mortality that may compromise physiological processes such as growth (Meesters, Noordeloos & Bak, 1994) and reproductive success (Van Veghel & Bak, 1994). Predation can also be associated with negative indirect effects; if tissue regeneration is incomplete or too slow following predation, spatial competitors such as algae and sponges can colonize the dead areas, potentially introducing a myriad of negative interactions (Bak & Steward-Van Es, 1980). Additionally, some corallivores are known to vector disease (Sussman et al., 2003; Williams & Miller, 2005), and tissue lesions (likely including those from partial predation) have been cited as a pre-requisite to disease transmission in Acropora cervicornis (Gignoux-Wolfsohn, Marks & Vollmer, 2012).

While predation among healthy coral populations can promote a stable equilibrium, within degraded coral populations, the per capita relative impact of corallivory is likely to increase as coral cover decreases, potentially affecting the fitness and recovery of affected coral populations (Jayewardene, Donahue & Birkeland, 2009). This was observed on surveyed reefs in Jamaica where populations of A. cervicornis were reduced by the acute disturbance of Hurricane Allen. As a result, predation by the corallivorous gastropod, Coralliophila abbreviata, was concentrated on remaining individuals resulting in further population declines of A. cervicornis rather than recovery (Knowlton, Lang & Keller, 1990).

Abiotic and biotic disturbances such as hurricanes, disease and bleaching not only reduce coral populations, but may affect the ‘condition’ of the remaining colonies yielding broken branches, lesions, compromised immune response (Bak & Criens, 1981), growth (Bak, 1983) and reproduction (Rinkevich & Loya, 1979). Furthermore, studies have shown that stressed corals are more susceptible to ambient predation (Morton, Blackmore & Kwok, 2002; Wolf & Nugues, 2013) suggesting that predation pressure on a focal coral may increase as a result of disturbance-related physiological stress in addition to a potential numerical effect (i.e., higher ratio of predators to surviving prey). Field studies on the impact of predation following disturbances such as Knowlton, Lang & Keller (1990) cannot distinguish the relative influence of diminished coral abundance versus coral condition.

The Acropora-Coralliophila relationship provides a good model for evaluating the relative influence of coral condition (i.e., healthy, diseased, damaged, etc.) on predator behavior. Considering the region-wide declines of Acropora spp., many populations may exist below a threshold abundance where they are now at heightened vulnerability to predation pressure. Coralliophila abbreviata is an obligate corallivore with an asymmetrical preference for Acropora spp. prey (Johnston & Miller, 2014) and may thus pose a substantive risk to the recovery of decimated Acropora populations. In the Florida Keys, predation by C. abbreviata was the most prevalent factor causing tissue loss among remnant populations of A. palmata, accounting for approximately 29% of all tissue mortality (Williams & Miller, 2012), and is recognized as one of the top three proximal threats to the recovery of wild (Bruckner, 2002) and restored Acropora populations (Johnson et al., 2011) throughout the Caribbean. Currently, management actions are underway throughout the Caribbean to enhance Acropora population recovery via conservation and restocking strategies. Understanding the interactions of factors contributing to the loss of Caribbean acroporids may allow for more effective conservation strategies for these species. We took advantage of the experimental system offered by in situ nursery propagation of A. cervicornis to conduct field choice assays testing the hypothesis that susceptibility to colonization by the corallivore, C. abbreviata, would be enhanced for corals with disease or mechanical damage relative to an apparently healthy coral.

Materials and Methods

This study was conducted at a coral field nursery operated by the Coral Restoration Foundation (CRF) located off Tavernier, Florida (24°59′N, 80°26′W) from June to September 2013 (under permit # FKNMS-2013-065). The nursery is located on sand bottom surrounded by seagrass at a depth of 10 m and provided Acropora cervicornis fragments for use in choice experiments. Coralliophila abbreviata were collected from Pickles Reef (24°59′N, 80°24′W), transported 2.2 km to the nursery and kept in holding cages until use in choice experiments. Snails were held separately based on the host species from which they were collected: A. cervicornis (n = 38) or A. palmata (n = 55). Snails were starved for at least one week prior to use in choice trials.

Treatment chambers were made by modifying plastic compartment boxes (40 × 23 × 8 cm). Each chamber contained two choice arenas with a removable lid, which was secured with cable ties during trial periods (Fig. 1A). A single choice arena consisted of a Y-maze with the subject snail staged at one end facing two treatment lanes and a treatment coral loosely secured at the end of each lane. Rectangular cutouts covered with window-screen mesh were at each end of the treatment chamber to facilitate flow. Chambers were oriented relative to predominant currents to allow water flow from the treatment coral toward the snail staging area. To minimize water exchange between treatment lanes and arenas, Velcro felt was glued to the top of the walls of each treatment lane allowing the lid to lay flush with no gaps. Each of ten experimental chambers was secured to a cinder block and set on sand bottom approximately 1 m apart (Fig. 1B).

Three types of prey choice experiments were conducted (Figs. 1C–1E): healthy coral versus diseased coral (H v D, n = 52), healthy coral versus coral with mechanical damage (H v M, n = 58) and healthy coral versus coral with active snail predation (H v P; n = 3). Prior to the start of the experiment, healthy fragments were snipped from apparently healthy nursery colonies and the cut surfaces were allowed to heal for at least two weeks in the field in order to yield completely undisturbed, healthy tissue. Disease progression is intermittent in A. cervicornis requiring actively diseased samples to be identified immediately at the start of a trial. Thus, diseased branch tips were collected from cultured coral with active tissue loss and snipped immediately prior to trials within an area of already-dead skeleton (i.e., no tissue disturbance), approximately 2 cm from the tissue margin. Fragments with mechanical damage were prepared by snipping branch tips from healthy coral, and, immediately prior to the start of a trial, a 3–5 cm long abrasion was created with a clean, dead A. cervicornis branch. This treatment was included to mimic the type of damage inflicted from abrasion during storms or hurricanes. At the end of any trial, if a snail was actively feeding on a healthy coral, the coral and feeding snail were immediately transferred to serve as a choice in a subsequent H v P trial (Fig. 1E).

Coralliophila abbreviata are known to be gregarious (Bruckner, Bruckner & Williams, 1997); so there is an aspect of prey choice that may be purely social. Thus, we also tested snail preference between a healthy coral fragment versus a solitary snail with no coral (H v S, n = 45). In these trials, a treatment snail was tethered via a small length of twine glued to its shell at the treatment end of one of the Y-maze lanes with a healthy coral at the end of the other lane (Fig. S1).

The position of the healthy coral and treatment coral/snail was alternated between the two lanes in subsequent trials. Following each trial, the chamber was flushed of all sand and debris, and the walls and floors were rigorously wiped down with a brush or a diver’s gloved-hand to reduce carryover of mucus or other potential cues. Trials were pooled among those conducted during daytime (8:00 am to approximately 4:00 pm; ∼8 h duration) and nighttime (sunset to a few hours after sunrise; ∼16 h duration).

During each trial, subject snails were left alone to choose a treatment lane. If the subject snail did not travel more than 5 cm down a treatment lane or remained in the staging area, the trial was determined as ‘no choice’ and excluded from analyses (n represents only the trials where a choice was made; Table 1; see Table S1 for daytime and nighttime trials separated). The proportion of trials in which a choice was made (i.e., response rate) ranged from 41 to 72% across the three treatments (D, M and S), which is well within the range of response rates reported in other published Y-maze choice studies using gastropod subjects (range: 27–100%; Nakashima, 1995; Avila, 1998; Rilov, Gasith & Benayahu, 2002).

Table 1 Summary of Y-maze trial treatments.

Paired choice experiments testing prey preferences by Coralliophila abbreviata. N gives the number of successful trials (for the treatment paired with H) with subject snails from each of two host corals, Acropora cervicornis (Ac) or A. palmata (Ap). The ‘# of no choice trials’ represents additional trials conducted wherein the subject snail did not make a choice.

	Treatment	Origin	N	# of no choice trials	
			Ac	Ap	Sum		
	Healthy coral (H)	5 cm branch tip snipped from nearby stock colony and allowed to heal for 2 weeks	–	–	–	–	
	Diseased coral (D)	4–9 cm branch tip with active disease snipped from nearby stock colony immediately prior to deployment in trial. Breaks were made on dead skeleton approximately 2 cm below active disease margin	29	23	52	32	
VERSUS	Mechanically damaged coral (M)	Healthy branch tip with 3–5 cm section actively abraded with a dead A. cervicornis branch immediately prior to deployment in a trial	32	26	58	23	
	Solitary snail (S)	Snail tethered at end of one treatment lane with no coral	28	17	45	65	
	Coral with active snail predation (P)	Snail feeding on a healthy fragment from an immediately prior trial	2	1	3	0	

For each treatment except H v S, trials among the two host-source subject snails are pooled for analysis as no difference was found between snails sourced from A. palmata and A. cervicornis. Additionally, trials are pooled among daytime and nighttime for each treatment as no difference was found in preferences expressed during daytime versus nighttime. Differences in frequencies of choices made between healthy and treatment corals/snail were assessed using a Pearson’s chi-squared test. The variation in activity levels (i.e., proportion of trials in which a choice was made) between daytime versus nighttime trials was analyzed using a 2 × 2 contingency table (Statistica Statistical Software v6.0).

Results and Discussion

Coral condition significantly affected prey preference of Coralliophila abbreviata snails sourced from Acropora spp. host colonies. In 70.8% of the trials, snails preferred corals with either disease (df = 1, p < 0.001, Fig. 2A) or mechanical damage (df = 1, p = 0.01, Fig. 2A) over apparently healthy corals. Other studies have shown similar results for the Pacific corallivorous snail, Drupella rugosa, where snails were attracted to corals stressed by either mechanical damage, low salinity or low water temperature suggesting that corals stressed by additional factors beyond the scope of the present study may manifest a similar enhanced susceptibility to corallivores (Morton, Blackmore & Kwok, 2002; Tsang & Ang, 2015).

It is well known that chemoreception is important in foraging behavior of marine benthic organisms (Kohn, 1961; Hay, 2009). Although the specific mode of attractant to prey has not been studied for C. abbreviata, it seems likely that variable chemical cues may underlie their preferences. Abraded coral tissue releases mucus and interstitial content that contains primary metabolites such as proteins and amino acids which may attract consumers (Hay, 2009). The release of mucus and/or secretions by damaged cells were the suggested cause of attractants for increases of the corallivorous snail, D. rugosa, to stressed corals in Hong Kong (Morton, Blackmore & Kwok, 2002). Similarly, Kita et al. (2005) showed an increase in ‘feeding-attractant activity’ by the corallivorous snail, Drupella cornus, when offered montiporic acids isolated from the prey coral, Montipora sp., which are suggested to be expelled with coral mucus. As this study only examined behavioral responses based on short distance cues, further studies should determine a range of distances that snails are able to detect such cues to better infer snail foraging patterns on a reef scale.

A similar mechanism that attracts snails to mechanically damaged corals may apply to diseased coral tissue as it may result in tissue deterioration and the production of excess mucus. However, although mechanically damaged corals used in this study appeared to produce more mucus and expel more interstitial content than diseased corals, snails had a slightly stronger attraction to diseased coral (75% of choices made) than mechanically damaged coral (65.5% of choices made) suggesting there may be something more complex attracting the snails than simple quantity of these exudates. Additionally, as this study did not induce disease for H v D treatment corals, it is possible that an undescribed physiological difference in corals making them more susceptible to disease may also make them more attractive to snails rather than the diseased condition per se.

Though there was only opportunity to conduct limited trials of the H v P treatment (n = 3), 100% elicited a choice and 100% chose the fragment under active predation (P) suggesting a strong bias toward prey fragments with active conspecific snail feeding. To ensure that the subject snail was not attracted to the mere presence of a conspecific, H v S trials were conducted and showed no significant preference for the conspecific snail relative to a healthy coral (Fig. 2B). However, the following of conspecific mucus trails has been reported for the marine mud snail, Ilyanassa obsolete (Trott & Dimock, 1978) and has been suggested to account for aggregation behavior in the corallivorous snail, Cyphoma gibbosum (Gerhart, 1986). Yet, considering that traces of previous mucus trails were removed (see methods), this preference may be the result of damaged coral tissue releasing interstitial attractants (described above) or feeding mucus produced by the feeding snail.

Intraspecific behavioral differences in C. abbreviata sourced from A. palmata and A. cervicornis colonies were evident in H v S trials (Fig. 2B) and in activity level based on time of day (Table S1). Snails sourced from A. cervicornis showed a strong preference for healthy coral over a solitary snail with no coral, while snails sourced from A. palmata showed no preference for either treatment. Additionally, snails sourced from A. palmata were significantly more active at night than during the day (evidenced by proportion of trials in which a choice was made), whereas, snails sourced from A. cervicornis showed no difference in activity level between daytime and nighttime trials. Intraspecific behavioral differences have been shown for a number of marine species (e.g., Stachowicz & Hay, 2000; Crosby & Reese, 2005; Jordaens, Dillen & Backeljau, 2009). In C. abbreviata, behavioral and population structural differences (e.g., size/age structure, sex ratio, etc.) have been previously documented between acroporid and non-acroporid host corals (Hayes, 1990; Baums, Miller & Szmant, 2003a; Johnston & Miller, 2007), despite genetic results showing it to be a single species throughout the Caribbean (Johnston, Miller & Baums, 2012). To our knowledge, no previous studies have documented intraspecific behavioral differences of C. abbreviata between two Acropora spp. host corals as evident in our results. One explanation may be that A. cervicornis-sourced snails were presented with their native host prey species, whereas A. palmata-sourced snails were presented with a non-host prey species. Some sort of host conditioning may result in differing attractiveness of native versus a congeneric prey alternative. However, such differences were unexpected since the tissue of A. cervicornis and A. palmata are qualitatively similar (thin tissues on perforate skeleton) and have similar nutritional quality as indicated by C:N ratios (Ac = 6.1 ± 0.9, Szmant, Ferrer & FitzGerald, 1990; Ap = 6.3 ± 0.3, Baums, Miller & Szmant, 2003b).

Coral disease and physical damage occur regularly on coral reefs, and, as changing climate is predicted to bring increases in intensity and/or frequency of strong storms (McWilliams et al., 2005; Knutson et al., 2010), as well as disease (Harvell et al., 2002), it is crucial to understand how corals may be directly and indirectly affected by these disturbances. This study highlights the enhanced vulnerability of remnant coral populations following acute disturbance events such as storms or disease outbreaks due to corallivore behavioral preferences. The indirect effect of attracting snail predators to these impacted corals implies ongoing tissue loss from predation inhibiting potential recovery. Furthermore, there is likely a complex feedback between disease risk and snail predation as C. abbreviata has been shown to vector disease among A. cervicornis colonies (Williams & Miller, 2005). Although corallivory by invertebrates is relatively well-documented, corallivore behavior and its potential influence on recovery of threatened or endangered coral populations has received little attention. Understanding such behavioral complexities can aid in epidemiological and predictive modelling of disease dynamics and transmission in Acropora spp. populations as well as improved species recovery strategies such as targeting snail removal efforts (Williams et al., 2014) following specific types of disturbance.

Figure 1 Photo examples of experimental Y-maze chamber design and treatment coral fragments.

(A) Experimental chamber design. Each chamber has two separate choice arenas, depicted as ‘1’ and ‘2.’ White circles depict the initial staging area for the subject snail. The treatment corals are attached at the far end of each treatment lane. (B) Cages were aligned in the same direction facing into the current. (C) Photo example of a treatment coral fragment with disease, (D) a fragment with mechanical damage and (E) a fragment with active snail predation.

Figure 2 Results of Y-maze trials.

Percent of successful choice assay trials comparing snail preference of (A) healthy (H; white bars) A. cervicornis fragments versus fragments with compromised condition (black bars; diseased [D] or mechanically damaged [M]) and (B) H versus a conspecific snail (S; grey bars) presented separately for subject snails sourced from A. cervicornis and A. palmata hosts. Asterisks indicate significant results (Pearson Chi-squared tests, p < 0.05). (The total number of successful trials is given at the base of each bar.)

Supplemental Information

Table S1 Comparison of number of choices made during daytime and nighttime trials

(A) The number of successful (i.e., a choice was made) and unsuccessful trials (i.e., no choice was made) conducted during daytime and nighttime for snails sourced from Acropora palmata (Ap) and A. cervicornis (Ac). (B) The percent of successful trials conducted during daytime and nighttime.

Click here for additional data file.

Figure S1 H v S trial

Photo example of a trial with a healthy coral fragment versus a solitary snail (H v S).

Click here for additional data file.

Data S1 Raw data

Click here for additional data file.

This study would not have been possible without the collaboration and facilities provided by K Nedimyer (Coral Restoration Foundation), to whom we are extremely grateful. We thank C Marmet, DE Williams, J Fisch and L Richter for their help in the field.

Additional Information and Declarations

Competing Interests

Author Contributions

Field Study Permissions

The authors declare there are no competing interests.

Allan J. Bright conceived and designed the experiments, performed the experiments, analyzed the data, wrote the paper, prepared figures and/or tables.

Caitlin M. Cameron performed the experiments, analyzed the data, reviewed drafts of the paper.

Margaret W. Miller conceived and designed the experiments, wrote the paper.

The following information was supplied relating to field study approvals (i.e., approving body and any reference numbers):

Florida Keys National Marine Sanctuary permit # FKNMS-2013-065.

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
