# Peer review of "Enhanced susceptibility to predation in corals of compromised condition"

_PeerJ, doi:10.7717/peerj.1239_

## Round 0.1 · original submission · Minor Revisions

Please, consider all the suggestions made by the reviewers in the revised version of your manuscript.

Reviewer 1 ·

Basic reporting

This article is clearly written and provides a good background to justify the study performed. I have personally observed this phenomenon of C. abbreviata congregating and feeding on dying coral tissue and find this study to be a useful contribution to the literature for those working with coral disease and conservation.The authors have performed three/four experiments to understand whether the corallivorous snails involved in this behavior are drawn to diseased versus damaged coral or to conspecifics.

Experimental design

The experimental design for the field component is sound, but it would be useful to clarify a couple of things.
Please describe the H v P trial and number of replicates in the methods (line 113).
As trials were pooled among those conducted during daytime and those conducted at night (lines 124-126), it would be useful to see these numbers separately in Table 1 or written in the methods.

There is a problem with the chi-square test, however. Looking at the data, although there were more than 50 replicates, this does not mean you have more than 50 degrees of freedom for the chi-square test. The df is calculated as the number of categories -1. So, for example in the H v D treatment, the df = 2-1, because there are two categories (diseased and healthy). Or if day/night are also included, then there are 4 categories and df = 3. Were daytime and nighttime categories included in this test? This should be explained in the methods.

Validity of the findings

Overall, the findings are reasonable based on the results. It is interesting that there appears to be a difference in how the snails respond to conspecifics based on the coral species that the snail was collected from (figure 2b). However, this is not mentioned at all.

Additional comments

Rephrase the sentence at lines 9-12 in the Abstract.

Reviewer 2 ·

Basic reporting

There are sections of the paper that are unclear. I have indicated those areas and suggested wording on the manuscript itself.

Experimental design

Please give the dimensions of the experimental chambers. Also, the H v P trials are never explained in the Methods section and need to be explained.

Validity of the findings

The findings appear valid based on the methods and results.

Additional comments

mucus- noun
mucous- adjective
This needs to be corrected throughout.

Annotated reviews are not available for download in order to protect the identity of reviewers who chose to remain anonymous.

---

## Round 0.2 · accepted · Accept

The manuscript has been revised and improved.